# A Decade of Wishes-Changes in Maternal Preference of the Mode of Delivery among Polish Women over the Last Decade

**DOI:** 10.3390/medicina57060572

**Published:** 2021-06-03

**Authors:** Agnieszka Jodzis, Maciej Walędziak, Krzysztof Czajkowski, Anna Różańska-Walędziak

**Affiliations:** 12nd Department of Obstetrics and Gynecology, Medical University of Warsaw, Karowa 2 St., 00-315 Warsaw, Poland; agnieszkajodzis@gmail.com (A.J.); krzysztof.czajkowski@wum.edu.pl (K.C.); aniaroza@tlen.pl (A.R.-W.); 2Department of General, Oncological, Metabolic and Thoracic Surgery, Military Institute of Medicine, 04-141 Warsaw, Poland

**Keywords:** cesarean delivery on maternal request, mode of delivery, maternal preference, shared decision making, vaginal birth after cesarean

## Abstract

*Background and Objectives:* The maternal preference of mode of delivery is an important problem in respect of patient’s autonomy and shared decision-making. The objective of the study was to obtain information about women’s preferences of the mode of delivery and knowledge about the cesarean section and its’ consequences. *Materials and Methods*: The study was based on a survey filled in by 1175 women in 2010 and 1033 women in 2020. Respondents were asked about their preference of mode of delivery, possible factors influencing their decision and their knowledge about risks and benefits of cesarean section. *Results:* There was a significant increase in the rate of women who declared cesarean section as their preferred mode of delivery, from 43.97% in 2010 to 56.03% in 2020 (*p* < 0.05). In 2010 26.51% of women thought that choice of mode of delivery should be their autonomic decision, 46.36% preferred decision-sharing with their obstetrician, 25.64% thought that cesarean section should be performed for medical indications only (respectively 34.86%, 44.45% and 19.38% in 2020). *Conclusions:* There has been a significant increase in the rate of Polish women who prefer cesarean delivery over the last decade, as well as in the rate of women who consider the mode of delivery as their autonomic decision.

## 1. Introduction

Patient-centered care requires shared decision making, which is a unique process in the context of choosing the mode of delivery [1]. The decision influences not only the mother, but also the baby, whose well-being depends on mother’s good decision-making. It is vital to provide adequate information about the vaginal delivery and cesarean section to allow women understand the real advantages and disadvantages, also including time of recovery after both ways of giving birth [2]. The relational context of the process of informing about the procedure followed by the women’s trust in their healthcare professional are known to be important factors [3]. Although some authors accept performing cesarean delivery on maternal request (CDMR), their obstetricians should always recommend against medically not indicated cesarean section, as a part of good medical practice [4]. The maternal preference of the mode of delivery can be very strong and the risk of postpartum depression and post-traumatic stress symptoms is increased in case a woman delivers in a different mode than she preferred, especially in psychologically vulnerable women [5,6,7]. The feeling of having control over the mode of delivery is very important to many women, although not necessarily guarantees them a positive birth experience [8]. There are various reasons for the maternal request of cesarean section, including fear of vaginal birth, fear of pain, feeling of autonomy, concern for fetal health, concern about future sexual life, fear of pelvic floor damage, previous or present pregnancy complications [9,10,11].

Over the last decade, the lifestyle and level of patient’s knowledge about medical procedures has changed due to global access to social media. That also includes women’s knowledge about the advantages and disadvantages of different modes of delivery and therefore we wanted to verify whether it has influenced their preference for having a cesarean section. 

The objective of the study was to obtain information about Polish women’s preferences of the mode of delivery and knowledge about the cesarean section and its’ consequences for the mother and baby. The secondary was to compare the state of their knowledge now and before a decade.

## 2. Materials and Methods

This study was designed as an anonymous online and paper survey with the aim to collect data about the opinion of Polish women about the cesarean delivery. The online survey was created and distributed via social media in 2010 and 2020. The questionnaire was filled by anonymous female voluntary participants aged 18 or more years. The survey including the same questions was conducted in 2010 and 2020. Data was collected for 2 months in both time periods. We collected data from 1175 women in 2010 and from 1033 in 2020. Women were asked to fill in a questionnaire containing questions about their basic characteristics (age, place of residence, education, socioeconomic status, height, weight, comorbidities and obstetric history) and questions about their opinion and knowledge about cesarean section. The second part of questionnaire included questions about their preferred method of delivery, decision-making, cesarean delivery on maternal request, the level of pain and time of recovery after the operations and the impact of cesarean section on mother and baby, including possible complications. There were no exclusion criteria apart from not female gender, minority (less than 18 years old) and missing or conflicting data between the answers. Questionnaires including incorrect, missing or conflicting data were excluded from further analysis in the study. 

### 2.1. Statistical Analysis

Statistical analysis was performed using Statistica 13 (StatSoft. Inc., Tulsa, OK, USA). U-Mann Whitney test and t-student tests were used for quantitative data comparison as required. Two-sided Fisher’s exact test and chi-square test were used for categorical and binary data comparison as required. *p* value < 0.05 was considered significant.

### 2.2. Ethical Considerations

The study was anonymous, performed in accordance with the ethical standards aid down in the 1964 Declaration of Helsinki and its latter amendments (Fortaleza). Participants were informed about the aim of the study and informed consent was obtained electronically prior to the beginning of the survey. The approval from Warsaw Medical University Ethics Committee was obtained 19 March 2013 from code AKBE/21/13.

## 3. Results

The medium age of 2010 group was 28.0 (±8.8) years and of 2020 group 32.0 (±6.7) years old, the vast majority of the two groups having been of reproductive age (95% in 2010 vs. 98% in 2020). There were no differences between the groups in the place of habitation, about 68% of respondents from both 2010 and 2020 group lived in cities of more than 50,000 habitants. 10.01% of 2010 group vs. 8.20% of 2020 group had medical education and respectively 49.45% vs. 63.65% higher education. 93.73% respondents from 2010 admitted medium or high socioeconomic status vs. 98.60% in 2020. 75.02% women from 2010 vs. 71.37% from 2020 declared no comorbidities, 13.23% of all respondents suffered from thyroid malfunction. 

The baseline characteristics of the groups is presented in Table 1.

18.25% of respondents from 2010 were pregnant at the moment of filling in the questionnaire vs. 9.03% from 2020. Out of pregnant respondents, in 2010 14.67% were in the 1st trimester, 22.67% in the 2nd trimester and 62.67% in the 3rd trimester, respectively 22.68%, 35.05% and 42.27% in 2020. 

51.44% of respondents from 2010 had a history of previous pregnancy vs. 77.12% from 2020. 15.82% of women on 2010 admitted a history of miscarriage (11.64% had one miscarriage, 2.76%–2, 1.42%–3 or more) vs. 19.98% women in 2020 (respectively 15.33%, 3.62%, 1.02%).

Out of 530 women from 2010 who had a history of delivery, 33.77% had had a cesarean section vs. 53.82% of 810 women from 2020 who had a history of delivery. Less than a half women in both groups had a history of more than one delivery (41.56% in 2010 vs. 48.84% in 2020). Among women, who had a history of vaginal delivery 42.35% in 2010 and 57.65% in 2020 declared it was a difficult delivery.

In 2010 26.51% of women thought that it was only their right to decide about the method of delivery, 46.36% wanted decision-sharing with their obstetrician, 25.64% thought that cesarean section should be performed for medical indications only (respectively 34.86%, 44.45% and 19.38% in 2020), (Figure 1). 

The differences between women’s opinions present a statistically significant difference (*p* < 0.05), the wish for personal choice of mode of delivery having risen from 25.64% in 2010 to 34.86% in 2020. Only 29% of respondents both in 2010 and 2020 were aware that cesarean section may have negative influence on the newborn, 7.39% in 2010 and 5.52% in 2020 thought the influence on the newborn was positive, 28.17% vs. 29.93% declared it had no influence and 35% in both groups admitted no knowledge on the subject. Only 32.11% of respondents in 2010 vs. 45.98% in 2020 thought a cesarean delivery had influence on breastfeeding. 40.07% in 2010 and 30.01% in 2020 thought that cesarean section was not associated with pain, compared to 36.22% vs. 51.72% who considered it a painful experience.

Among respondents who answered the question whether they would have decided to have a cesarean section without medical indications, 47.83% in 2010 and 37.48% in 2020 answered positively. 

In 2010 41.14% respondents accepted the possibility of additional charge for CDMR, compared with 45.15% in 2020 and there was a significant decrease in the rate of women against (42.42% vs. 35.91%; *p* = 0.006). 

There was a statistically significant increase in the rate of women who declared cesarean section as their preferred mode of delivery, from 43.97% in 2010 to 56.03% in 2020 (*p* < 0.05) The decrease in the rate of vaginal delivery as the preferred mode of delivery was even higher, 57.95% in 2010 vs. 42.05% in 2020, due to higher rate of non-decided respondents (11.35% vs. 19.14%).

The structure of the preferences is presented in Figure 2.

## 4. Discussion

The results of our study show that maternal preference of the mode of delivery in Poland has changed over the last decade, the rate of preference for the cesarean section having had increased from almost 44% of women in 2010 to more than 56% in 2020. The rate of preference of cesarean section in Poland has always been high compared to other countries (i.e., 3.1% in an American study from Pennsylvania from 2019 [1]. Our results also present an increasing trend in the rate of performed cesarean sections, with 34% of women with a history of delivery by cesarean section in 2010 up to 54% in 2020. This trend may be a result of changes in clinical management, maternal preference that creates pressure on obstetricians in decision-making process and a lower threshold for deciding about an operative delivery among obstetricians [12]. Obstetric associations in Europe, Canada and United States do not recommend CDMR [13]. The rate of obstetricians accepting CDMRs differs between the countries. The most supportive of CDMRs are American (84.5% in the state of Maine) and Australian (77.3%) obstetricians, followed by 57.9% in Italy and 53% in Turkey. The other end is represented by 23% of Canadian, 15% of Spanish and 14% of Chinese obstetricians [14,15]. The Canadian Society of Gynaecologists and Obstetricians emphasizes the importance of appropriate counseling for the woman about the risks and benefits of cesarean section without medical indications so that she makes an informed decision. If the patient still insists on CDMR, they can either perform the cesarean section after 39 + 0 weeks of gestation or refer the patient to ask for a second opinion [16]. The Polish Society of Gynecologist and Obstetricians do not have presently a standpoint about CDMR. National Polish recommendations about cesarean section from 2008 were strongly against CDMR and underlined the role of the obstetricians as the only and finally responsible for choosing the method of delivery [17]. However, the latest Polish recommendations from 2018 that replaced those from 2008 or Polish do not include information about official attitude towards CDMR [18]. There are neither studies about the personal opinion of Polish obstetricians about CDMR or the rate of CDMRs in Poland. The subject of CDMR remains a taboo as most obstetricians continue prohibiting CDMR according to previous recommendations. Polish maternity care is based on public hospitals and the role of private hospitals in Poland, where CDMRs are more accessible remains negligible. Additionally, there is no reliable data from private hospitals. The new recommendations about cesarean section include more indications to perform a repetitive cesarean section compared to the previous ones and creating a higher number of elective cesarean deliveries in the second term-pregnancy, all of whom would have a definite operative delivery in third and higher number term-pregnancies. This is one of the most important reasons for the increasing rate of cesareans in Poland [18].

The increasing rate of cesarean sections leads to a vicious circle as it is followed by a low rate of successful vaginal births after a cesarean (VBAC) [19]. There different studies trying to find an optimum process of counseling or patient-centered decision support tools to increase the rate of trials of labor after a cesarean section [20,21]. The determination of a pregnant woman in attempting VBAC was found to be one of the most important factors, together with adequate information received from the obstetrician, participating in prenatal courses and family influence. Higher education was associated with an increased rate of CDMR [22]. 

However, there are a few recent studies which have shown that CDMR may have better outcomes for the mother and baby at least in the short term. In a recent study, Guo et al. compared the risk of maternal and neonatal adverse outcomes in a group of Canadian women in low-risk pregnancies who had a planned CDMR (0.4% of the group, n = 1827 women) and those who had a planned vaginal delivery. The factors that influenced the decisions about CDMR were found to be late maternal age, higher educational level, being White, nulliparity, higher than recommended gestational weight gain, in vitro fertilization and delivery at a hospital at higher level of perinatal care. The researchers found that planned CDMR was associated with a decreased risk of both maternal and neonatal adverse outcomes. Adjusted RR was 0.41 for any maternal components and 0.42 for any neonatal component. CDMR compared to planned vaginal delivery was associated with a lower risk of newborn death, birth trauma (≥2000 g), Neonatal Intensive Care Unit (NICU) admission and 5-min Apgar score <7 [13]. These findings are of very high interest, although further studies on the impact of CDMR on long-term outcomes, including breast-feeding and children’s risk of respiratory illness and infections should be conducted.

We observed a rising trend in the number of women who claimed it was their autonomic right to choose the mode of delivery, the rate having increased from more than 25% in 2010 to almost 35% in 2020. The rate of women who prefer co-deciding with their obstetricians is comparable between the 2010 and 2020 groups (46% vs. 44%). Shared decision making gives women a feeling of autonomy, and reliable information about risks and benefits of cesarean delivery from the health professional is a very important factor contributing to a woman’s decision [23,24]. Woman’s decision about the mode of delivery is determined by many influences, including social interactions, family, friends and the media, emotional experiences and previous experience of childbirth [25].

Karlstrom et al. compared the satisfaction of childbirth experience among women who preferred and had a cesarean section and those who preferred and had a vaginal birth. The cesarean group included 34 women and the vaginal delivery group included 659 women, all of whom filled in two questionnaires during their pregnancy and one two months after giving birth. The results of the study were that women from the cesarean group experienced higher level of fear of the childbirth than women from the vaginal delivery group and were not satisfied with the antenatal care and decision making process. Additionally, they had a more negative birth experience and had doubts about having more children [2].

### Limitations of the Study

The possible limitation of our study can be the recall bias and the subjectivity of patients’ opinion. Another limitation was that the survey was conducted mostly among women who were able to fill it by means of internet and therefore distribution via social media excluded the possibility of direct control of the respondents or calculation of the response rate. However, there was no incentive to introduce dishonesty into responses. 

## 5. Conclusions

There has been a significant increase in the rate of Polish women who prefer cesarean delivery over the last decade. There also has been an increase in the rate of women who consider the choice of mode of delivery as their autonomic right, with the rate of women preferring co-deciding with their obstetrician having remained stable over the years. Factors influencing those changes will be subject to our further studies.

## Figures and Tables

**Figure 1 medicina-57-00572-f001:**
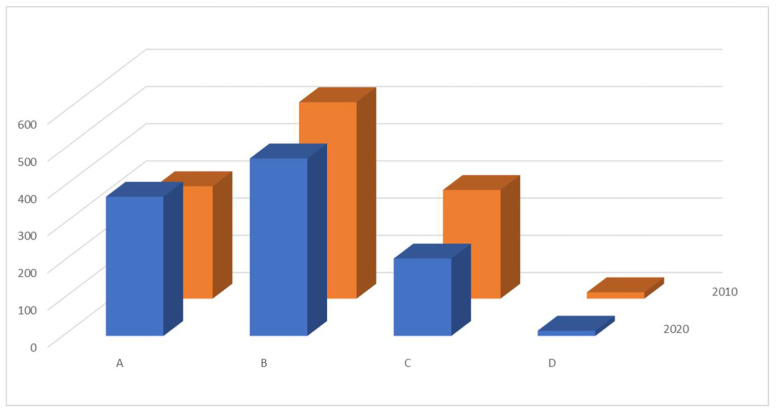
Decision about mode of delivery–respondents preference. A–woman’s autonomic decision B–co-decision with the obstetrician C–cesarean delivery only in case of medical indications D–non-decided respondents.

**Figure 2 medicina-57-00572-f002:**
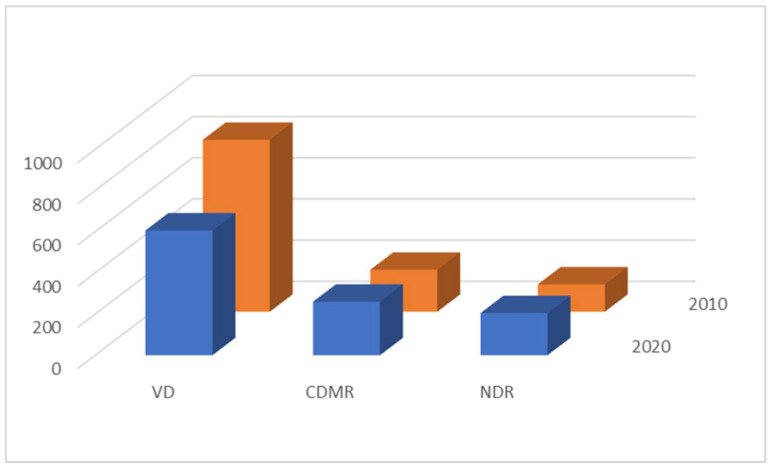
Structure of the preferences. VD–vaginal delivery, CDMR–cesarean delivery on maternal request, NDR–non-decided respondents.

**Table 1 medicina-57-00572-t001:** Basic characteristics and comorbidities of the study group.

Variable	2010	2020
Participants, *n*	1175	1033
Mean age, years (±)	28.0 (±8.8)	32.0 (±6.7)
Place of habitation:		
cities >100,000	663 (56.23%)	613 (57.08%)
cities 50,000–10,000	139 (11.79%)	120 (11.17%)
cities <50,000	166 (14.08%)	146 (13.59%)
village	211 (17.90%)	195 (18.16%)
Education:		
primary education	53 (4.50%)	19 (1.77%)
secondary education	425 (36.05%)	283 (26.37%)
higher	583 (49.45%)	683 (63.65%)
medical professional	118 (10.01%)	88 (8.20%)
Socioeconomical status:		
low	73 (6.27%)	15 (1.40%)
medium	944 (81.03%)	792 (73.95%)
high	148(12.70%)	264(24.65%)
Co-morbidities:		
none	828 (75.27%)	725 (71.15%)
1	222(20.18%)	256 (25.12%)
2	39 (3.55%)	34 (3.34%)
3	11 (1.00%)	4 (0.39%)

## Data Availability

The data presented in this study are available on request from the corresponding author. The data are not publicly available.

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
