# Peer review of "A Decade of Wishes-Changes in Maternal Preference of the Mode of Delivery among Polish Women over the Last Decade"

_medicina, 2021, doi:10.3390/medicina57060572_

Round 1
Reviewer 1 Report
I read with extreme interest this article about women's birth modality expectations and preferences. However, this study seems to be a qualitative research article that needs to follow the appropriate recommendation for data reporting in qualitative studies. Therefore, please follow the SRQR guideline: O'Brien BC, Harris IB, Beckman TJ, Reed DA, Cook DA. Standards for reporting qualitative research: a synthesis of recommendations. Acad Med. 2014 Sep;89(9):1245-51. doi: 10.1097/ACM.0000000000000388. PMID: 24979285.
In addition, I would like to stress some further points:
1) In the background section of the abstract, the aims of the study are missing; please amend.
2) The methods should be improved according to the SRQR guideline.
3) In table 1, in the four-last line, you wrote "non," but I suppose you mean "none."
4) At the end of the limitations paragraph, please change ".." to "."
Author Response
Thank you for your review and important remarks. We changed the manuscript accordingly.
We changed the methods section to include all the aspects included in the SRQR guideline.
We added the aims of the study to the background section of the abstract.
We corrected table one and the limitations paragraph.
Reviewer 2 Report
This is a simple straightforward and well conducted research on the preferences of Polish women regarding the mode of delivery.
The clause “A decade of wishes” in the title is not very meaningful, but does make it attractive, unless the authors can think of a better alternative.
The introduction and discussion is fairly comprehensive, nuanced and informative.
Page 6, Lines 186-189: The meaning of the first sentence in this paragraph is not clear and hence confusion. A better sentence would be, “However, there are a few recent studies which have shown that CDMR may have better outcomes for the mother and baby at least in the short term.”
Limitations of the study have been well described.
A very brief discussion on some more factors could be included as follows,
- Effect of higher education and socioeconomic status
- Trend towards having fewer babies (and hence CS preferred).
- Attitudes by Obstetricians to caesarean while counselling the women
- organisation of maternity care. Private vs public. Role and influence of Nurse-midwives.
- Confidence of women in the maternity services.
Author Response
Thank you for positive review and your remarks.
We tried to reformulate the title.
We changed the first sentence on page 6 according to your suggestion.
We added the factors you suggested to the discussion, including the obstetricians attitude, official recommendations, organization of maternity care, public and private hospitals.
Thank you for mentioning the problem of higher education and socioeconomic study, we consider it a very important issue and all the factors influencing women’s preference of mode of delivery will be analyzed in our next study.
We could not analyze the trend towards having fewer babies nor the problem of women’s confidence as unfortunately the questionnaire did not include adequate questions.
The role of nurse-midwives in Polish obstetrics is limited to assisting obstetricians during women’s visits throughout the pregnancy, taking care of patients during hospitalization and delivery. Women have contact with midwives mostly when attending pre-delivery courses, but it is estimated that only up to 10% of pregnant women attend those courses so the main counseling role is in the hands of obstetricians.
Reviewer 3 Report
The authors raised a very interesting social problem concerning changes in the preferences of the way of delivery over the last ten years among Polish women. They showed an increased interest in the Caesarean section as the preferred delivery route and an increased willingness to make autonomous decisions about the delivery method. The authors on representative study groups proved the trends observed in everyday clinical practice in recent years. The factors influencing the preferences of the delivery route that the authors promised to identify in future studies seem to be of particular interest.
Congratulations to the authors of an interesting work, waiting for further publications on this topic.
Author Response
Thank you for your review and positive opinion about our work. We are in train of preparing a new manuscript about the factors influencing women’s preferences of the delivery route and we are hoping to be able to present it in the nearest future.
Round 2
Reviewer 1 Report
I found the manuscript significantly improved and suitable for publication in its present form.